# Enhancing shift current response via virtual multiband transitions
Sihan Chen [1,2] ✉, Swati Chaudhary[3,4,5] ✉, Gil Refael[2,6] ✉ & Cyprian Lewandowski [7,8] ✉

Materials exhibiting a significant shift current response could potentially outperform conventional solar cell materials. The myriad of factors governing shift-current response, however, poses significant challenges in finding such strong shift-current materials. Here we propose a general design principle that exploits inter-orbital mixing to excite virtual multiband transitions in materials with multiple flat bands to achieve an enhanced shift current response. We further relate this design principle to maximizing Wannier function spread as expressed through the formalism of quantum geometry. We demonstrate the viability of our design using a 1D stacked Rice-Mele model. Furthermore, we consider a concrete material realization - alternating angle twisted multilayer graphene (TMG) - a natural platform to experimentally realize such an effect. We identify a set of twist angles at which the shift current response is maximized via virtual transitions for each multilayer graphene and highlight the importance of TMG as a promising material to achieve an enhanced shift current response at terahertz frequencies. Our proposed mechanism also applies to other 2D systems and can serve as a guiding principle for designing multiband systems that exhibit an enhanced shift current response.

The bulk photovoltaic effect (BPVE) is debated as a promising alternative source of photocurrent to conventional p-n junction-based photovoltaics[1–8]. One of the microscopic mechanisms behind the BPVE is the shift current, which generates direct current upon electromagnetic radiation in non-centrosymmetric materials[2,3]. Intuitively, the physical origins of shift current trace back to a real-space shift experienced by an electron wavepacket upon an interband excitation driven by linearly polarized light[9,10]. Shift current is a second-order optical response whose contribution arises from direct optical transitions and transitions via a virtual state[2,11–13]. Multiple factors like interband velocity matrix elements, density of states, number of bands, band gap as well as the quantum geometry[2,13–18] determine the magnitude of the shift current response. The goal of establishing guiding principles to maximize the shift current response is an ongoing problem with direct technological ramifications[11,19–21].

In multiband systems, virtual transitions through intermediary bands are an additional transition process that can contribute to the magnitude of nonlinear optical response. In addition to a direct transition from the initial band to the final band, virtual transition scatters through an intermediary band. This manifests in the structure of the shift current response, which can be decomposed into direct and virtual transition components[2,12]. Previously, the seminal work[9] outlined the design principles to maximize the shift current from direct transitions. This work was inspired by ferroelectrics and orthorhombic monochalcogenides[22–24] (materials with large band edge responsivities which exhibit BPVE in the visible and far-infrared range). For the two-band model under consideration in the Ref. [9] work, the virtual transitions are not present, and thus, the focus of ref. [9] lies primarily on maximizing singularities in the joint density of states (JDOS). Conversely, in the case of materials with multiple bands, for example alternating angle moiré graphene[25–27] (see Fig. 1a–d for a typical bandstructure), it becomes imperative to explore an alternative approach to maximizing shift current response (e.g., refs. [1,28]), in particular based on the significance of contributions stemming from virtual transitions.

In this paper, we demonstrate a viable mechanism that leverages virtual transitions in multiband systems to significantly enhance shift current response, see Fig. 1e. The enhancement is achieved by designing materials with multiple bands close in energy, thereby increasing the number of possible virtual transitions through the increased number of intermediary bands. Increasing the number of closely spaced energy bands, however, can carry implications also for the quantum texture of the electron states, as exemplified by the Berry connection sum rules[29]. We demonstrate that shift

[1]Kadanoff Center for Theoretical Physics, University of Chicago, Chicago, IL, 60637, USA. [2]Department of Physics, California Institute of Technology, Pasadena, CA, 91125, USA. [3]Department of Physics, The University of Texas at Austin, Austin, TX, 78712, USA. [4]Department of Physics, Northeastern University, Boston, MA, 02115, USA. [5]Department of Physics, Massachusetts Institute of Technology, Cambridge, MA, 02139, USA. [6]Institute for Quantum Information and Matter, California Institute of Technology, Pasadena, CA, 91125, USA. [7]National High Magnetic Field Laboratory, Tallahassee, Florida, FL, 32310, USA. [8]Department of Physics, Florida State University, Tallahassee, FL, 32306, USA. ✉e-mail: sihanc@uchicago.edu; swati.chaudhary@austin.utexas.edu; refael@caltech.edu; clewandowski@magnet.fsu.edu

**Fig. 1 | Bandstructure of T2G, T5G, T7G, T8G, and shift current conductivity of T5G with varying external displacement field strength.** Here *TNG* denotes the bandstructure of an *N*-layer TMG. **a–d** Bandstructure of TMG system with external displacement field strength $U = 20$ meV at physical twist angles $\theta = 0.8°, 1.32°, 1.63°, 1.71°$ for $N = 2, 5, 7, 8$, respectively. **e** Shift current conductivity for T5G at $\theta = 1.32°$ with varying external displacement field strength. The result of increased external displacement field leads to band mixing, which leads to enhancement of shift current via virtual transition as well as a new peak at low frequency. Non-vanishing signal as $\omega \rightarrow 0$ in this figure is a consequence of finite Lorentzian broadening (See Supplementary Note 10 for more details).

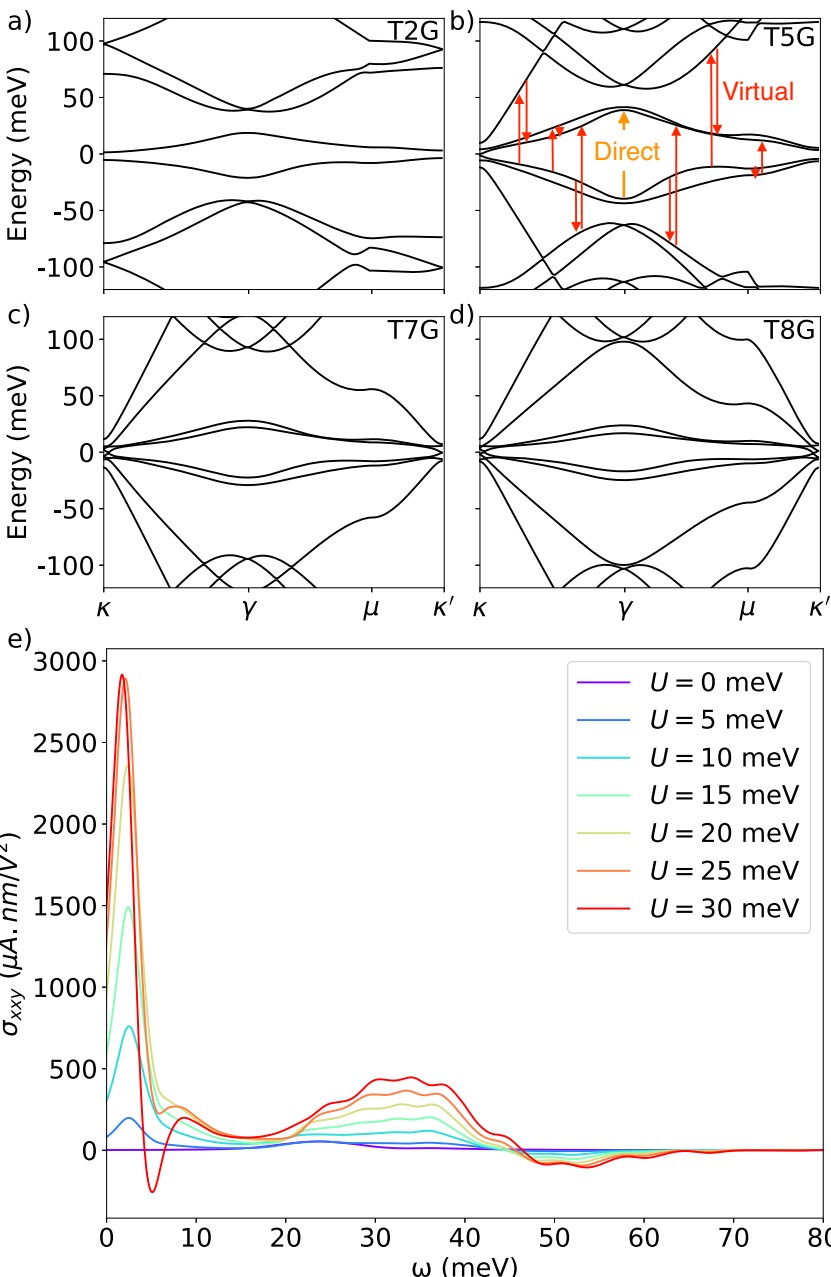

current, whose magnitude is partly controlled by such quantum geometric matrix elements (e.g., refs. 16,30,31), can take advantage of this mechanism. Specifically, we find that the transition rate between the initial and final bands involved in the photo-absorption process is enhanced in the regime where the wavefunction of the involved states (including the virtual state) becomes more delocalized.

These two design principles are the key results of our paper, which underlying physical mechanisms we elucidate first using a 1D multi-chain Rice-Mele model, and then we focus on the multilayer moiré graphene materals[25–27]. Moiré materials such as twisted multilayer graphene (TMG) have acquired much attention due to their multiple flat bands exhibiting nontrivial topology and exotic phenomena such as superconductivity and correlated-insulating behavior[32–34]. Our focus on the moiré materials stems from the fact that they are expected to host a large shift in current response[30,31], but we emphasize that the general principles we discuss apply to other materials well.

## Methods

### Theory of shift current

Shift current is a second-order DC response to linearly polarized AC fields[2,12]. It is characterized by a third-rank conductivity tensor[2] given by

$$\mathbf{J}^\beta = 2\sigma^\beta_{\alpha\alpha}(0, -\omega, \omega)E^\alpha(\omega)E^\alpha(-\omega) \qquad (1)$$

where $\mathbf{J}^\beta$ is the $\beta$-th component of current density, $E^\alpha(\omega)$ is the electric field with frequency $\omega$ in the $\alpha$ direction, with $\alpha, \beta$ denotes spatial indices $x, y$. The shift-current conductivity[2,9,12] is given by the expression:

$$\sigma^\beta_{\alpha\alpha}(0, -\omega, \omega) = \frac{\pi e^3}{\hbar^2} \sum_{m,n} \int d^2\mathbf{k} f_{mn} \mathbf{S}^{\beta\alpha}_{mn} |\mathbf{A}^\alpha_{mn}|^2 \delta(\omega - \varepsilon_{mn}), \qquad (2)$$

where it has the form of a shift vector $\mathbf{S}^{\beta\alpha}_{mn} = \mathbf{A}^\beta_{mm} - \mathbf{A}^\beta_{nn} + \partial_{k_\beta} \mathrm{Arg}(\mathbf{A}^\alpha_{mn})$ that characterizes the shift in the localization of Bloch wavefunction upon

transition from state $|u_n\rangle$ to state $|u_m\rangle$, weighted by the transition amplitude given by the interband Berry connection $\mathbf{A}_{mn}^\alpha = i\langle u_m|\partial_{k_\alpha} u_n\rangle$. The sum is over all energy bands, where $\varepsilon_{mn} = \varepsilon_m - \varepsilon_n$ is the energy difference between two states, and $f_{mn} = f_m - f_n$ is the difference in occupancy of their energy levels.

To visualize the transition process and draw a distinction between direct and virtual transitions, we use generalized sum rules to replace wavefunction derivatives with sums over all states of matrix element derivatives[2,11,12]. The shift current integrand defined as $R_{mn}^{\alpha\alpha\beta} = |\mathbf{A}_{mn}^\alpha|^2 \mathbf{S}_{mn}^{\beta\alpha}$ is given by,

$$
R_{mn}^{\alpha\alpha\beta} = \frac{1}{\varepsilon_{mn}^2} \mathrm{Im}\left[ \frac{h_{mn}^\alpha h_{nm}^\beta \Delta_{mn}^\alpha}{\varepsilon_{mn}} + w_{mn}^{\alpha\beta} h_{nm}^\alpha \right]
$$
$$
+ \frac{1}{\varepsilon_{mn}^2} \mathrm{Im}\left[ \sum_{l \neq mn} \left( \frac{h_{nm}^\alpha h_{ml}^\beta h_{ln}^\alpha}{\varepsilon_{ml}} - \frac{h_{nm}^\alpha h_{ml}^\alpha h_{ln}^\beta}{\varepsilon_{ln}} \right) \right]. \quad (3)
$$

where $h_{mn}^\alpha = \langle m|\partial_\alpha H|n\rangle$, $\Delta_{mn}^\alpha = h_{mm}^\alpha - h_{nn}^\alpha$, and $w_{mn}^{\alpha\beta} = \langle m|\partial_{k_\alpha}\partial_{k_\beta} H|n\rangle$ (See Supplementary Note 1 for derivation). The first term represents a direct transition from band $n$ to band $m$, and the second term represents virtual transitions through an intermediary band[12]. In a two-band system in 1D, only $w_{mn}^{\alpha\beta}$ term contributes as the first term in the direct transition term becomes purely real and virtual transitions are not present in the two-band model. In the TMG system, when we expand momentum to linear order near the $\kappa/\kappa'$ points, $w_{mn}^{\alpha\beta}$ vanishes. Figure 1b shows a schematic depiction of direct and virtual transitions between different bands in TMG. The virtual transition sums over all intermediary bands, and we expect such a process to be the dominant contribution in systems containing multiple bands of similar energies. The coupling between these bands can be controlled by the external displacement field which can be used to enhance the shift current conductivity as shown in Fig. 1e. In addition to discerning the direct and virtual transition contributions, Eq. (3) is numerically more amenable as it avoids dealing with the issue of gauge fixing when evaluating the derivative of the wavefunction directly[11,12]. We stress, however, that while our analysis, and indeed Eq. (3), are all carried out in the velocity gauge, our conclusions are gauge independent, i.e., equally present if the calculation was done in the length-gauge. In Supplementary Note 1 we further discuss the equivalence of the two gauges.

## Results
### Multilayer Rice-Mele model
In order to study the role of virtual transitions in controlling the magnitude of the shift current response, we construct a toy model with multiple close-spaced bands. Specifically, we consider a 1D stacked Rice-Mele (RM) model to demonstrate the interplay between virtual transitions and bandstructures in multilayer systems. The Rice-Mele model is a prototypical model for one-dimensional ferroelectrics[9] represented by the Hamiltonian

$$
H = \sum_i \left[ \left( \frac{t}{2} + (-1)^i \frac{\delta}{2} \right)(c_{i+1}^\dagger c_i + \mathrm{h.c.}) + (-1)^i \Delta c_i^\dagger c_i \right], \quad (4)
$$

where $t$ is the hopping potential, $\delta$ parameterizes the difference in hopping strength between the two neighboring sites with unit cell length $a$, and $\Delta$ is the staggered onsite potential. The canonical Bloch Hamiltonian in momentum space is given by ref. 35

$$
H(k) = \sigma_x t \cos\frac{ka}{2} - \sigma_y \delta \sin\frac{ka}{2} + \sigma_z \Delta. \quad (5)
$$

To construct a multilayer Rice-Mele model, we denote the dependence of the momentum-space Bloch Hamiltonian on its parameters as $H_i = H_i(t_i, \delta_i, \Delta_i)$ where $i$ is the layer index, and with the full Hamiltonian for $N$ layers taking the following form:

$$
H_{\mathrm{MRM},0}(k) = H_1(k) \oplus H_2(k) \oplus \cdots \oplus H_N(k) \quad (6)
$$

As written above in Eq. (6), the multilayer Rice-Mele Hamiltonian is block-diagonal thus transitions between the different blocks are impossible. To enable virtual transitions between the RM sectors belonging to different layers we introduce couplings between layers. Specifically, consider the case of $N = 2$ with band mixing, the two-layer Rice-Mele (2RM) model has Hamiltonian of the form

$$
H_{\mathrm{2RM}}(k) = \begin{pmatrix} H_1(k) & \varepsilon V \\ \varepsilon V^\dagger & H_2(k) \end{pmatrix}, \quad (7)
$$

where $V$ is the operator that characterizes interactions and $\varepsilon$ parameterizes the mixing strength. Figure 2a shows the bandstructure of the 2RM model with $V = \sigma_x$, and $(t, \delta_1, \Delta_1, \delta_2, \Delta_2) = (1, 0.8, 0.7, 0.86, 0.6)$ with (without) mixing as depicted by solid (dashed) line. For this particular form of coupling, the two low (high) energy bands hybridize, which widens the gap between black and red in the low (high) energy sector. This effect manifests itself most strongly at $ka = \pm\pi$, where the bands flatten out significantly in comparison to the unmixed case as shown in Fig. 2a. Here, we fixed $t$ to be the same for both RM models and vary the hopping amplitude by tuning $\delta_i$.

When mixing is added, transitions can occur between all four bands, allowing virtual transitions to enhance the overall shift current response. As shown in Fig. 2b, the transition from valence to conduction band of the first RM model is enhanced after band mixing due to virtual transitions through a different RM sector. The peaks in the response are determined by the energy gaps at $ka = \pm\pi$, 0, where the joint density of the state is the largest. Level repulsion as a result of band mixing widens the gap, consequently shifting the location of the peak to a slightly higher frequency. In addition to the selective enhancement at precise frequencies, the overall shift current conductivity $\sigma(\omega)$ over the whole frequency range is increased. Specifically, using a figure of merit

$$
M = \left| \int \sigma d\omega \right| \quad (8)
$$

as a measure of the overall shift current response[28], we find that $M_{\mathrm{unmix}} \approx 9$ and $M_{\mathrm{mix}} \approx 13.5$, demonstrating an enhanced response.

The coupling-induced hybridization of bands not only flattens the dispersion around $ka = \pm\pi$ but also engenders additional transition pairs throughout the Brillouin zone that further enhance the overall shift current response. The total shift current conductivity is now obtained by summing over all possible transition pairs, which include transitions from one RM sector to another. The sign of conductivity depends on the specific transition pairs. On the basis of quantity $M$, introduced in Eq. (8) above, some transition pairs will contribute towards the cancellation of conductivity. Nevertheless, an overall enhancement is still observed compared to models without band mixing (See Supplementary Note 5 for more details). Different transition pairs give rise to peaks in conductivity at different frequency ranges. To avoid cancellation due to opposite signs, one could work at specific frequencies to excite the corresponding transition pairs.

Furthermore, by studying the shift current response of 2RM model at filling $v = 0.5$, where the chemical potential lies in between the band edge of the top two bands, see Fig. 2c, we find an additional shift current response between the two bands 2 and 3 with $M_{32} \approx 42.4$ in the low-frequency range as shown in Fig. 2d. This response relies on band mixing and is much larger than the response at charge neutrality considered in the previous paragraph and in Fig. 2b. This response is dominated by direct transition as virtual transitions are suppressed by the large energy gap. Only one peak occurs at a frequency determined by the energy gap difference at the band edge because the states around $k = 0$ do not contribute to transitions. The reasons for such a large response are twofold: small energy gap and large joint density of state, which can be inferred from Eq. (3) and Eq. (2), respectively. The exclusion of states around $k = 0$ would not drastically reduce the shift current response, because the gap is much larger at $k = 0$ in comparison to the gap at the band edge.

**Fig. 2 | Bandstructure and shift current conductivity of the two-layer Rice-Mele model and its dependence on model parameters $\delta_2$.**
**a** Bandstructure of 2RM model without band mixing (dashed line) and with band mixing of $\varepsilon = 0.1$ (solid line). The red/black band refers to the first/second RM model, with intracell hopping strength $v_i = \frac{1}{2}(t_i + \delta_i)$, intercell hopping strength $w_i = \frac{1}{2}(t_i - \delta_i)$, and sublattice offset potential $\Delta_i$, where $i = 1, 2$ indexing the different RM layers. A schematic description of the model is included, with $\varepsilon$ coupling the different sublattice degrees of freedom between the chain. The parameters of 2RM are $(t, \delta_1, \Delta_1, \delta_2, \Delta_2) = (1, 0.8, 0.7, 0.86, 0.6)$. **b** Shift current conductivity due to transitions between the red band (first RM model) with and without band mixing. The presence of mixing enhances the peak due to the transition at the band edge via virtual transitions through the second RM bands. **c** Bandstructure of the top two bands, with the orange dashed line denoting chemical potential at filling $\nu = 0.5$. **d** Shift current conductivity due to transitions from band 2 to band 3, showing a large response at a low frequency determined by the energy gap of band 2 and 3 at $ka = \pm\pi$.

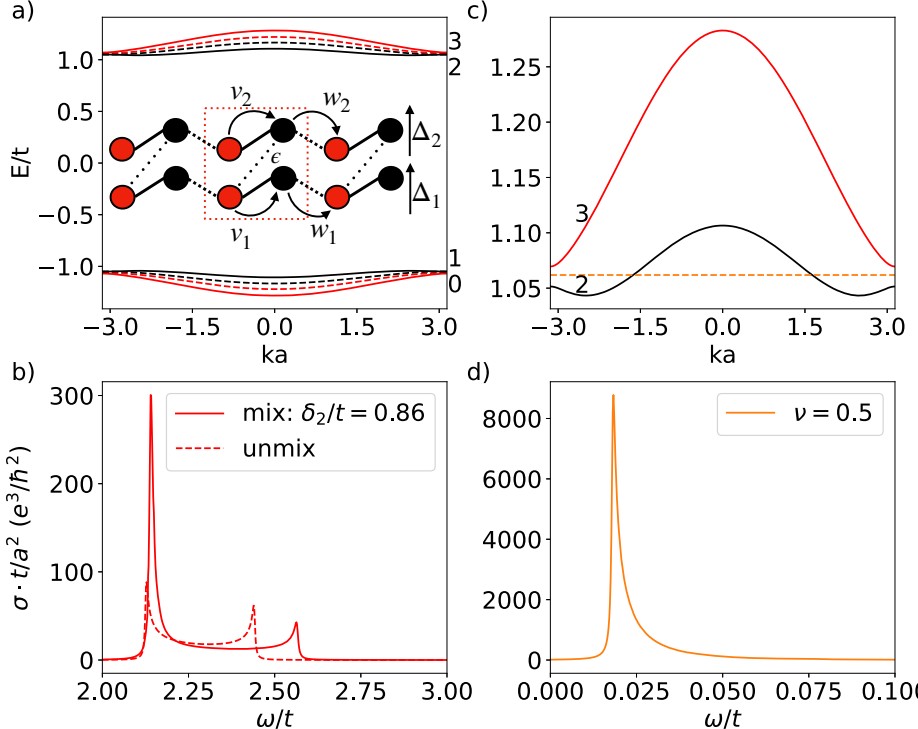

We can understand the above results at $\nu = 0.5$ with the help of the ref. [28]. As pointed out in ref. [28], the energy gap is not the only criteria which dictates the shift current response but band dispersion also plays an important role. In a typical tight-binding two-band model, the structure of the valence band fully determines the structure of the conduction band. This unique relationship defines an upper limit on quantity $M$ in terms of the energy gap, bandwidth, and the range of hopping as discussed in ref. [28]. These three factors also dictate the extent of delocalization of the wavefunction in a two-band model which was highlighted[28] to be the most important criteria for enhanced shift current response. On the other hand, for a system with energetically close multiple bands, there are additional physical principles that control the limit of shift current response. In particular, many recent works have demonstrated how quantum geometry which manifests in multiband systems can contribute to physical quantities, for example such as the superfluid stiffness, beyond their nominal limits inherent in single-band or two-band models[34,36–39].

While the multiband aspect of our system manifests explicitly through virtual transition contributions to shift-current response, the specific choice of model parameter is crucial in achieving enhanced response via virtual transitions. Figure 3a compares the magnitude of total shift current response $M$ with and without band mixing of transition from band 0 to band 3 fixing $\delta_1, \Delta_1, \Delta_2, \varepsilon$, and vary $\delta_2$, breaking down the contribution to total response into direct and virtual transition. We see that the effect of band mixing can lead to a reduction in direct transition contribution to shift current. Simultaneously, however, there can be a significant enhancement in the virtual transition contributions to the shift current.

Specifically, there exists a parameter regime, where this increase in the virtual transition component is enough to offset the reduction in direct transition. In this region, where we picked $\delta_2 \approx 0.86$, we observe an enhanced shift vector, $S_{30}(k)$, near the band edge, as shown in Fig. 3b, c. We attribute this enhancement effect to the sudden increase in the spatial delocalization of Wannier wavefunctions stemming from strong band hybridization at those band parameter values.

This Wannier function delocalization can be studied more systematically with the help of the Fubini-Study (FS) metric[36,40]. For a single band

$m$, the FS is defined by

$$g_{\mu\nu}^m(k) = \text{Re}[G_{\mu\nu(k)}^m], \quad (9)$$

with

$$G_{\mu\nu}^m(k) = \langle \partial_\mu m(k)|(1 - |m(k)\rangle\langle m(k)|)|\partial_\nu m(k)\rangle. \quad (10)$$

The FS metric defines a distance between momentum states, and its integral over the Brillouin zone serves as a lower bound for Wannier function localization[39,41,42] (See Supplementary Note 2 for more details). Figure 3d shows a sweep across the region of $\delta_2$ that exhibits enhancement in total shift current via virtual transitions, and we find a corresponding increase in the FS metric integral $\int g^m(k)dk$, which implies a delocalization of the Wannier function. This observation is in agreement with previous works that demonstrate a correspondence between larger delocalization and larger shift vector[1], and consequently large shift current response[43]. The blue dashed line denotes the maximum of the integrated FS metric and where $S_{30}$ from Fig. 3c diverges at the band edge. The blue dashed line is where the band gap of the top/bottom is minimized, and it also corresponds to optical zero ($A_{mn}(k) = 0$). Hence, we identify a competition between the increasing of $S_{mn}(k)$ and decreasing of $|A_{mn}(k)|$, leading to the maximum of shift-vector integrand $R_{mn}$ defined in Eq. (3) occurring at the parameter near optical zeros (marked by orange dashed line), where the shift vector is still large, see Fig. 3d–f. This relation between maximized FS and shift current similarly holds in 2D, which is the procedure used to determine the optimal twist angle in TMG discussed in the next section.

We further note that the enhancement in FS metric $g(k)$ can be described by a Lorentzian function. (See Supplementary Note 3 for more details). This suggests that the sudden increase in FS metric, and consequently the spread of Wannier wavefunction, can be thought of as resonance in parameter space. It is this resonantly enhanced delocalization of Wannier wavefunction that leads to the enhanced shift current response. In momentum space, this enhancement is due to virtual transition via intermediary bands. Specifically, the enhancement comes from the small energy

**Fig. 3 | Parameter choice that leads to enhanced shift current response. a** Total shift current $M$ broken down into direct and virtual contributions. **b** Shift vector before and **c** after band mixing, demonstrating contribution mainly coming from band edge. **d** FS metric $g_{xx}^m(k)$ of band $m = 3$, **e** shift current integrand, and **f** shift vector at $ka = -\pi$, showing that enhanced response corresponds to a larger spread in the Wannier wavefunction. The dashed blue and orange lines denote the maximum of the FS metric integral and shift current integrand respectively.

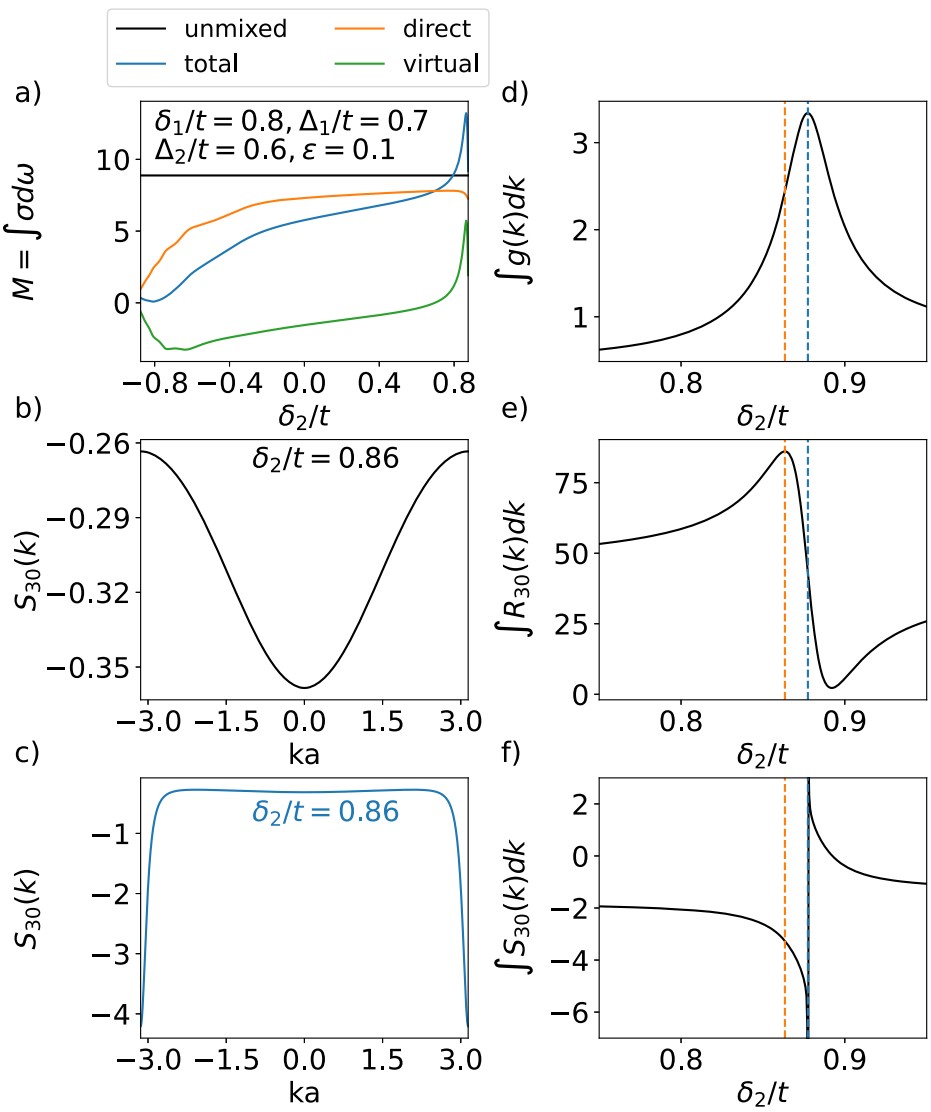

gap between the top two bands $E_{32}$ and the large matrix element $\text{Im}\{v_{20}\}$ with a pole occurring at $\delta_2 = \sqrt{\delta_1^2 + \Delta_1^2 - \Delta_2^2}$ at the band edge. (See Supplementary Note 4 for more details).

The proposed mechanism above suggests to stack multiple 1D systems together and induce band mixing. By carefully tuning the system's parameters, we find a region of parameters where the Wannier function becomes maximally delocalized. Although our analysis is constrained to the simple 1D model, we propose that the identified design principles can extend to more realistic models. Specifically, in the following section, we demonstrate these principles with the help of a realistic material model.

## Twisted multilayer graphene

Twisted bilayer graphene (TBG) has been proposed to exhibit a large shift current response due to its nontrivial flat-band topology[30,31]. We thus propose to use TBG as a viable building block to construct the above proposed multilayer system. Specifically, we will follow the stacking procedure introduced in ref. 25, that has been recently realized experimentally[26,27]. The single-particle spectrum of TBG is described by an effective continuum model (e.g., ref. 44). Consider Hamiltonian in the sublattice basis ($A_1, B_1, A_2, B_2$),

$$H_\theta = \begin{pmatrix} H_1 & U^\dagger \\ U & H_2 \end{pmatrix}. \tag{11}$$

Let $l = 1, 2$ denote the bottom and top layer, respectively. The intralayer Hamiltonian centered at the Dirac cone $\mathbf{K}_\xi^{(l)}$ given by

$$H_l = -\hbar v \left[ R(\pm \theta/2) \left( \mathbf{k} - \mathbf{K}_\xi^{(l)} \right) \right] \cdot (\xi\sigma_x, \sigma_y) + \Delta_l \sigma_z \tag{12}$$

where $\theta$ is the twist angle relative to the origin, $\pm$ corresponds to $l = 1/2$, and $\xi = \pm 1$ labels the valley index of the original graphene Dirac cone. We include the sublattice offset term $\Delta_l \sigma_z$ to break inversion $C_{2z}$ symmetry, which is essential for nonzero shift current response[7]. Experimentally, this is achieved by coupling TBG to the top/bottom layer substrate. The effective interlayer coupling $U$ has the form

$$U = \begin{pmatrix} U_{A_2,A_1} & U_{A_2,B_1} \\ U_{B_2,A_1} & U_{B_2,B_1} \end{pmatrix} \tag{13}$$

$$= \begin{pmatrix} u & u' \\ u' & u \end{pmatrix} + \begin{pmatrix} u & u'w^{-\xi} \\ u'w^{\xi} & u \end{pmatrix} e^{i\xi \mathbf{G}_1^M \cdot \mathbf{r}} \tag{14}$$

$$+ \begin{pmatrix} u & u'w^{\xi} \\ u'w^{-\xi} & u \end{pmatrix} e^{i\xi(\mathbf{G}_1^M + \mathbf{G}_2^M) \cdot \mathbf{r}}, \tag{15}$$

where $w = e^{i2\pi/3}$, and $\mathbf{G} = n_1\mathbf{G}_1^M + n_2\mathbf{G}_2^M$ is the moiré reciprocal lattice vector, for $n_1, n_2 \in \mathbb{Z}$. We choose $\mathbf{G}_i^M = R(-\theta/2)\mathbf{G}_i - R(\theta/2)\mathbf{G}_i$, with the $\mathbf{G}_1 = (2\pi/a)(1, -1/\sqrt{3})$ and $\mathbf{G}_2 = (2\pi/a)(0, 2/\sqrt{3})$. Following the work done in ref. 30, we choose $u' = 90$ meV and $u = 0.4u'$, sublattice offset $\Delta = 5$ meV, and set $\hbar v/a = 2.1354$ eV and $a = 0.246$ nm; however, the qualitative behavior of the shift current that will be discussed below does not depend on these specific parameters of the model.

We alternatively twist each layer of graphene by an angle $\theta/2$ with respect to each adjacent layer to extend the bilayer to a multilayer system[25-27]. Coupling is only considered between adjacent layers. The Hamiltonian in the layer basis is given by

$$H^{(n)} = \begin{pmatrix} H_1 & U^\dagger & 0 & 0 & \cdots \\ U & H_2 & U & 0 & \cdots \\ 0 & U^\dagger & H_1 & U^\dagger & \cdots \\ \vdots & \vdots & \vdots & \vdots & \ddots \end{pmatrix} \qquad (16)$$

where we treated tunneling terms between all layers to be the same, and the twist angle of $H^{(n)}$ is given by $\theta_n = 2\cos(\frac{\pi}{n+1})\theta_{TBG}$. In the multilayer model, the sublattice offset term is added at the top and bottom layers to incorporate the effect of coupling to the hBN substrate. This multilayer Hamiltonian can be block-diagonalized into a direct sum of effective TBG pairs under a unitary transformation[25]. To induce virtual transition among different Hamiltonian layers, we consider applying an external displacement field on the top and bottom gates. (See Supplementary Note 6 for more details) The bandstructure of $N = 2, 5, 7, 8$ is shown in 1a–d, where we see $\lfloor n/2 \rfloor$ number of TBG-like bands with an additional Dirac cone if $n$ is odd. Bandstructure of T5G is reproduced in Fig. 4a with an external displacement field, including a schematic description of flat-to-flat band transition and flat-to-dispersive band transition. We study the transition between the lowest flat bands (colored in red) and observe an enhancement in shift current via virtual transitions to nearby flat bands once the external displacement field is included. To note, our TMG model has $C_{3z}$ symmetry, and by group theoretical method we can show that only two components of the conductivity tensor are independent (See Supplementary Note 9 for more details).

In our analysis, we neglect the role of electron-electron interactions, as our goal is to demonstrate just the principle of the physical mechanism that causes the shift current enhancement. The role of interactions in TBG and moiré materials is extensive as studied both theoretically (e.g., refs. 45–59) and experimentally (e.g., refs. 60–72). However, the exchange-driven physics, particularly at high temperatures ($T > 10$ K), should not affect the

qualitative ideas discussed in this manuscript. Moreover, as argued in Ref. 59, interactions can, in fact, give rise to a self-generated displacement field, which will induce hybridization between different energy sectors even in the absence of an external field. For simplicity of the discussion in what follows, we focus on the non-interacting model, leaving a detailed study of interactions on the shift-current photoresponse of a multilayer moiré system to future works.

The physical twist angle in TMG determines the effective twist angles of TBG-like bands, which in turn controls the energy difference between two flat band pairs in each energy sector of the model. The magnitude and direction of virtual transition are determined by the velocity matrix element and the energy gap of bands involved in the optical transition process as implied by Eq. (3). To receive maximum enhancement from virtual transitions, we choose twist angles such that the energy gap between two flat bands is as small as possible while the velocity matrix element is nonzero. The interplay between the velocity matrix element and energy gap determines the optimal twist angle for each multilayer system. Crucially here, since our goal is not to achieve a flat band, but rather to obtain closely spaced energy bands, the relevant twist angles differ from the magic angles of these multilayer systems. For example, from our numerics, we found $\theta_5 \approx 1.28°$ for T5G (TMG with $N = 5$ layers) helps optimize the shift current response, while the magic angle for T5G occurs at $\theta_{5,m} = 1.73°$, where a perfect flat band emerges. We outline the general procedure to determine the optimal twist angle in Supplementary Note 8 for a larger number of layers $n$, but for the next few $n$ values, these angles are given by: $\theta_6 \approx 1.47°$, $\theta_7 \approx 1.58°$, $\theta_8 \approx 1.68°$, $\theta_9 \approx 1.74°$. Correspondingly, there is an increase in shift current response due to reduced band gap and large joint density of states, an enhancement in direct transition. At the magic angle, the strongly interacting nature of the system requires us to incorporate Coulomb interaction, which enhances the resonance at non-interacting characteristic frequency due to filling-dependent band renormalization[30]. In TBG away from the magic angle, the shift current response is qualitatively the same, scales according to band gap and moiré length. In T5G, however, an enhancement via virtual transition to nearby flat bands occurs at angles where the two flat bands become close in energy, in addition to a new resonance peak at low frequency.

In contrast to the simple four-band model (the 2RM model) described in the last section, the TMG system has a contribution to shift current conductivity from both direct transition from flat to flat bands and virtual transitions from flat to dispersive bands coming from both the same and different TBG-like sectors. In the absence of an external displacement field, it is the virtual transition to dispersive bands within the same TBG sector that dominates the shift current response. With an external displacement field,

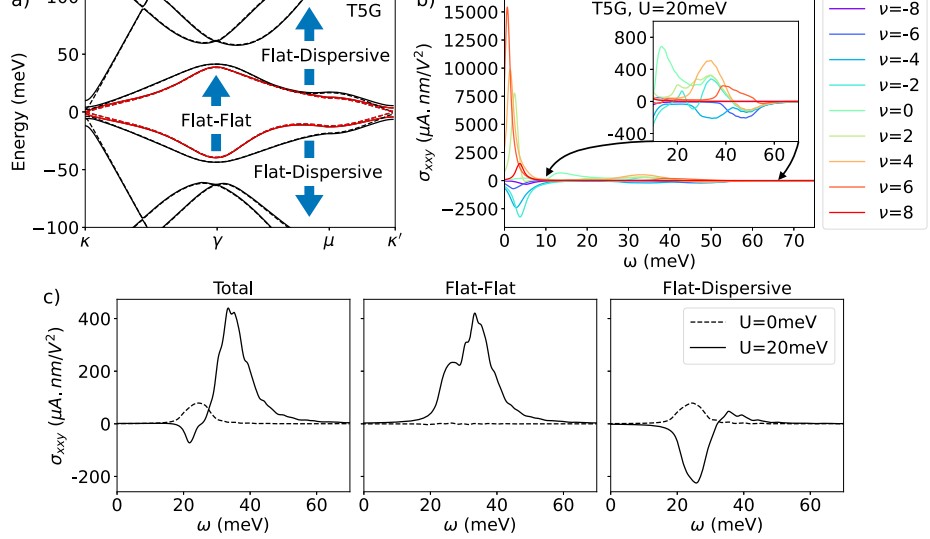

**Fig. 4 | Bandstructure of T5G and its total shift current conductivity plot at various filling, and breakdown of the enhancement effect via virtual transition of two flat bands at filling $v = 2$ electrons per moiré unit cell. a** A schematic depiction of the transition between flat to flat bands and from flat to dispersive bands, as well as the bandstructure of T5G with external displacement field $U = 20$ meV (solid line) and with $U = 0$ meV (dashed line) at $\theta = 1.32°$. **b** Shift current conductivity at various system fillings, demonstrating the ability to optimize conductivity at different frequencies by tuning the system filling. The inset shows a magnified portion of the whole plot. **c** Shift current conductivity due to transitions between bands colored in red in (**a**) at $v = 2$ (See Supplementary Note 7 for other fillings). Here we breakdown the total conductivity into virtual transition through flat bands and virtual transition through dispersive bands. Virtual transitions enhance both flat-flat and flat-dispersive contribution to shift current.

band mixing enables transitions to access different TBG-like sectors in the form of virtual transitions to either flat or dispersive bands of the other TBG-like sectors. Shift current conductivity $\sigma_{xxy}$ at $\nu = 2$ is shown in Fig. 4c. After band mixing, virtual transition through intermediary bands significantly enhances the response, in both flat-to-flat and flat-to-dispersive bands.

Summing up, all transition pairs reveal a giant peak at low frequency in shift current conductivity due to transitions to nearby flat bands and Dirac cone around the $\kappa$ points. As shown in Fig. 4b for various fillings at $U = 20$ meV, the peak of the response occurs at a frequency determined by the energy gap of those bands at the $\kappa$ points, roughly determined by the sublattice offset term $\Delta$. Previous work[73] also pointed out the effect of low-frequency divergence in topological semi-metals, where they attributed the divergence to singularity in quantum geometry. At a given filling, in addition to the giant peak at low frequency, there is a secondary peak occurring at a frequency proportional to the energy gap of the two flat bands at $\mu$, which corresponds to transitions among the flat band pairs. The appearance of two distinct peaks is a result of summing over different transition pairs. More explicitly, transitions between top/bottom pairs of flat bands give rise to low-frequency divergence, whereas transitions from bottom to top flat bands give rise to the secondary peaks. This is another unique feature of the TMG system compared to TBG system studied in previous works[30,31].

## Discussion

Our work proposes new design principles for optimizing the shift-current response in multiband systems. In particular, we demonstrated a mechanism based on the enhancement of virtual transition contributions in systems with multiple layers. The enhancement arises mainly from the mixing of bands belonging to different layers. The mechanism is explained using a 1D toy model where the stacking order provides an additional knob to tune the shift-current response.

We have shown that in TMG systems, shift current response is increased by virtual transitions that occur between two types of bands that arise in multilayer moiré systems: nearly-degenerate nearby flat bands from different energy sectors and dispersive bands present in all sectors. Virtual transitions enhance the resonance at the original peaks, whereas transitioning among nearby bands around $\kappa$ point produces another peak in the conductivity at low frequency. Both enhancement phenomena occur over a wide range of fillings, with different fillings corresponding to a slightly different enhanced peak in conductivity. The effect of virtual transitions found in this paper relies on the existence of multiple flat bands and applies when the number of layers $N \geq 5$. Compared to TBG, T3G only has an additional Dirac cone at the $\kappa$ point, and although T4G has two flat band pairs, the second pair is too dispersive for the enhancement effect to occur. This makes shift current response in TMG for $N \geq 5$ qualitatively different from that of $N < 5$. As we increase $N$, the existence of additional flat band pairs can lead to further enhancement via virtual transitions. The physical angle at which we would observe enhanced shift current response is also expected to increase, making it potentially easier to realize experimentally.

Much of the experimental attention in the studies of TMG has been focused on the system's behavior near magic angles (e.g., refs. 60–72), where superconductivity and correlated-insulating behavior are revealed. In TMG systems, in addition to the set of magic angles that give rise to superconductivity and correlated-insulating behavior, we find a different set of twist angles that will lead to a large shift in current response. For the next few layer $n$ values, these angles are given by $\theta_5 \approx 1.28°$, $\theta_6 \approx 1.47°$, $\theta_7 \approx 1.58°$, $\theta_8 \approx 1.68°$, $\theta_9 \approx 1.74°$, determined by identifying peaks (maxima) in the trace of FS as a function of $\theta$. (See Supplementary Note 8 for details of the procedure). Near these angles, we find enhanced virtual transitions via nearby bands, as well as more delocalized wavefunctions. We caution that while this design principle allows to optimize shift current by seeking twist angles where FS metric is maximized, it should be used in conjunction with techniques that confirm whether a particular twist angle configuration is energetically favored (e.g., see ref. 74). Furthermore, the multilayered structure of this platform also contributes to increased photoresponsivity which matches that of three-dimensional materials[75]. These findings make

TMG a promising platform for designing devices that produce giant photocurrents at terahertz frequency. In this paper, we focused specifically on alternating angle stacking of the multilayer graphene due to its experimental relevance and the fact that the electronic structure can be brought to a block-diagonal form yielding the simple interpretation of multiple TBG-like systems at different effective twist angles[25]. In future work, we could explore different twisting schemes using the guiding principle of maximizing the FS metric to fully characterize the parameter space for enhancing shift current response in TMG systems.

Finally, we believe that multilayer systems consisting of other 2D systems could also yield similarly large shift-current responses. Given a generic 1D or 2D system that already exhibits a large shift current response, that response could be further enhanced by a stacking scheme that induces neighboring energy bands and a virtual transition between them. Twisted stacking of layered materials such as TMD's would be a natural way to implement the design principle that we suggest.

Motivated by Eq. (3), we see that to achieve a large shift current, we want to minimize the band gap $\varepsilon_{mn}$ across which electrons are driven. Simply engineering bands with a small band gap, however, will also shift the frequency peak to small $\omega$, making the current susceptible to thermal fluctuations at finite temperature[73] (indeed the low $\omega$ peak present in Fig. 4b is unlikely to survive at temperatures exceeding $T > 10$ K). In addition, the derivation of shift current expression via perturbation theory assumes small $|E/\omega|$, leading to worries concerning the breakdown of perturbation theory in the small $\omega$ regime. Keeping the large band gap between bands $m$ and $n$ and engineering nearby bands $l$ close to the target band, with $\varepsilon_{ml}$ small, allows enhancing the shift current response through virtual transitions, while maintaining the resonance $\omega$ large. This large gap can potentially render the enhancement due to virtual transitions robust against thermal fluctuations, thereby making the experimental observation of this effect likely. Furthermore, the FS metric is inversely proportional to the energy separation of two bands. When a pair of bands with nonzero interband Berry connection have a small energy gap, it leads to a large FS metric, which also corresponds to a large localization length of Wannier wavefunction and a large shift vector. This confirms the intuition that a large shift current arises when there is a large shift in the center of charge of the wavefunction[9,10] in real space upon excitation to the conduction band. Recently, nonlinear optical response has been formulated in the language of quantum geometry[76], and it has been shown that shift current is related to the quantum geometric connection. From this perspective, the meticulous adjustment of model parameters positions the material in close proximity to the geometric singularity in the bandstructure, leading to divergence of the geometric quantities that govern the shift current response.

The intricate interplay between shift current and quantum geometry has enabled its use as a probe of quantum geometry and interactions in topological materials[30]. Here we show two examples of enhanced shift current response via virtual transition near the band closing point. In topological materials, gap closing is associated with topological phase transitions. It would be interesting to study the shift current response in noncentrosymmetric multiband topological materials near critical points to see if we observe a corresponding enhanced response. This could potentially allow us to use shift current as a probe of topological phase transitions.

One could also imagine our design principle to be utilized to enhance other optical responses that demonstrate a geometric origin. Another common second-order bulk photovoltaic response is the injection current, which is shown to be related to quantum geometric tensor[73,76]. Engineering bands near geometrical singularity might also lead to large injection current response.

Furthermore, quantum geometric effects are not limited to optical responses but also lead to many other exotic phenomena like flat-band superconductivity[37–39,77], undamped plasmons[78], non-reciprocity in plasmonics[79,80] and in Landau-Zener tunneling[81]. The mechanism illustrated here for enhancement of shift-current response using multiband systems would potentially be applicable in a wide variety of quantum geometric effects.

## Data availability

The data that support the findings of this study are available from the corresponding author upon reasonable request.

## Code availability

The code that supports the findings of this study is available from the corresponding author upon reasonable request.

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

## Acknowledgements

We thank Roshan Krishna Kumar for useful discussions and collaboration on a related project. S.C. acknowledges support from the Summer Undergraduate Research Fellowship at Caltech. Swati Chaudhary acknowledges support from the National Science Foundation through the Center for Dynamics and Control of Materials: an NSF MRSEC under Cooperative Agreement No. DMR-1720595. G.R. expresses gratitude for the support by the Simons Foundation, and the ARO MURI Grant No. W911NF-16-1-0361 and the Institute of Quantum Information and Matter. C.L. was supported by start-up funds from Florida State University and the National High Magnetic Field Laboratory. The National High Magnetic Field Laboratory is supported by the National Science Foundation through NSF/DMR-2128556 and the State of Florida.

## Author contributions

All authors contributed to developing the theory and writing the manuscript. S.C., S.C., G.R., and C.L. conceived of the presented idea. S.C. (Sihan Chen) performed the calculations supervised by S.C. (Swati Chaudhary) and C.L. All authors discussed the results and contributed to the final manuscript.

## Competing interests

The authors declare no competing interests.
