## [Peer Review File · Communications Physics]

Reviewers' comments:

Reviewer #1 (Remarks to the Author):

The authors demonstrate an approach for enhancing shift currents by tuning virtual transitions in stacked materials. This is related to the approach of maximizing Wannier function spread which has appeared in other papers. The significance of this manuscript is in showing how these concepts can be realized in an existing materials system, the twisted graphene multilayers. The results are easily generalizable to other layered systems, or in fact any material which has a large number of tuning parameters to control the energy gap between virtual transitions. This manuscript should appeal to many readers who work on such materials.

The conclusions are well supported by the data. Even though electronic structure is not calculated at a first-principles level, the demonstration that the virtual transitions are tunable by the stacking angle at the level of a continuum model is sufficient to support the point that the authors are making. However, there may be practical challenges to implementing the proposed scheme. The preferred rotation angle may not be energetically favorable (see for instance Phys. Rev. B 103, 115427). The authors may want to explain if the proposed shift current enhancements are sensitive to small deviations from the optimal rotation angle.

The authors mention that shift currents in small-gap materials are subject to thermal fluctuations. It should be mentioned that the reported giant shift current peak at low frequency (~ 1 meV in Fig. 4b) is also subject to the same issue. At large filling factors, TMG effectively becomes a small gap material. The giant shift current values that are shown in this figure may not actually be observable as a result.

There could be a similar effect even at charge neutrality for the virtual transitions. Even though the conduction bands are unoccupied when the gap is large enough, the energy of the virtual transitions could also be subject to thermal fluctuations through phonon displacement, etc. It is unclear if the scheme is practical if it rests on fine tuning very closely-spaced energy levels. It may be that the results are actually robust to thermal fluctuations because all that is required is a small energy difference between any two bands at any point in the Brillouin zone, but the point needs to be discussed further.

The value in the proposed scheme is in the large number of tunable parameters. The authors chose to investigate only 1 out of 5 possible interlayer twist angles. They may wish to comment on the possibility of obtaining even stronger enhancements by tuning all twist angles independently.

Reviewer #2 (Remarks to the Author):

This work claims to propose a general design principle for strong shift-current materials based on enhancing virtual transitions.

This claim sounds very interesting but is not supported enough by data. Also, the authors need to support their claim of proposing design principles with a more thorough and concrete analysis. The authors present many calculation results that show significant contributions originating from virtual transitions. However, I do not see a systematic control of the virtual-transition contributions. What is the dominant variable that controls the virtual part, while not affecting the direct-transition part? Moreover, while the velocity-gauge formula Eq. (3) contains virtual transitions, the same expression in the length gauge does not contain any virtual transition.

In conclusion, I do not recommend this paper for publication unless 1) the authors clearly present a tuning knob for the virtual-transition parts with data and 2) how can the virtual transition takes a distinguished role while the length-gauge formula does not distinguish virtual and direct transitions.

Reviewer #3 (Remarks to the Author):

In the manuscript, the authors discussed a mechanism that exploits virtual transitions in multiband systems to enhance shift current response. They found that the enhanced shift current response is achieved through increasing the number of possible virtual transitions and maximizing Wannier function spread that is related to Fubini-Study (FS) metric (or quantum geometry). They demonstrated the viability of the mechanism using a one-dimensional multi-chain Rice-Mele model. Furthermore, they proposed that a twisted multi-layer graphene with a proper twist angle exhibits the large shift current response.

As the authors mentioned in the paper, there is a direct connection between shift current response and FS metric in one dimension, and FS metric provides a lower bound for the spread of Wannier function. I have a few questions here.

- Could the authors show a similar connection in two dimensions?
- Related to the first question: dose large FS metric generally tend to exhibit large shift current response?

Overall, the paper is clearly and logically structured. I believe that the results will be beneficial for people who search for materials that exhibit large shift current response. Therefore, I recommend this paper to be published in Communications Physics.

Reviewer #1 (Remarks to the Author):

The authors demonstrate an approach for enhancing shift currents by tuning virtual transitions in stacked materials. This is related to the approach of maximizing Wannier function spread which has appeared in other papers. The significance of this manuscript is in showing how these concepts can be realized in an existing materials system, the twisted graphene multilayers. The results are easily generalizable to other layered systems, or in fact any material which has a large number of tuning parameters to control the energy gap between virtual transitions. This manuscript should appeal to many readers who work on such materials.

We thank the referee for their feedback and positive evaluation of our work.

The conclusions are well supported by the data. Even though electronic structure is not calculated at a first-principles level, the demonstration that the virtual transitions are tunable by the stacking angle at the level of a continuum model is sufficient to support the point that the authors are making. However, there may be practical challenges to implementing the proposed scheme. The preferred rotation angle may not be energetically favorable (see for instance Phys. Rev. B 103, 115427). The authors may want to explain if the proposed shift current enhancements are sensitive to small deviations from the optimal rotation angle.

We would like to thank the referee for bringing this work to our attention, which we now include a citation to in the paper. While our results do not sensitively depend on the proximity to the optimal angle (as presence of the multiple proximal energy bands in the electronic spectrum is a consequence of the stacking construction), indeed we anticipate the signal to peak at the optimal twist angle. In the Appendix we studied the variation of Fubini-Study metric with twist angle θ . We noticed that there are several peaks in the trace of the FS metric as a function of θ which can serve as a guide to choose different twist angles when combined with results from paper that the Referee pointed us towards with respect to choosing which twist angles are energetically optimal. We added a brief discussion on the Referee's question in the manuscript.

Changes to Manuscript: Added a reference to the paper and discussion of the sensitivity to small deviations from the optimal angle.

The authors mention that shift currents in small-gap materials are subject to thermal fluctuations. It should be mentioned that the reported giant shift current peak at low frequency (~ 1 meV in Fig. 4b) is also subject to the same issue. At large filling factors, TMG effectively becomes a small gap material. The giant shift current values that are shown in this figure may not actually be observable as a result.

We thank the Referee for this question. We agree with the referee that the small frequency response can be influenced by thermal fluctuations. However, the enhancement we have discussed is not limited to very small frequencies. In fact, the plots shown in Fig. 4c consider frequency (> 20 meV) and show a significant enhancement with displacement field. This enhancement arises because of hybridization between different virtual bands arising from the displacement field. We clarified discussion in the manuscript to reflect the Referee's question.

Changes to Manuscript: Clarified discussion of thermal effects.

There could be a similar effect even at charge neutrality for the virtual transitions. Even though the conduction bands are unoccupied when the gap is large enough, the energy of the virtual transitions could also be subject to thermal fluctuations through phonon displacement, etc. It is unclear if the scheme is practical if it rests on fine tuning very closely-spaced energy levels. It may be that the results are actually robust to thermal fluctuations because all that is required is a small energy difference between any two bands at any point in the Brillouin zone, but the point needs to be discussed further.

We thank the Referee for this question. Indeed the effect of virtual transitions enhancing the shift current response is present at charge neutrality point as well, please see Fig. 4b and a new figure in the supplement Fig. S3. This enhancement of shift current by virtual transitions is not an effect of fine-tuning - it naturally appears when there are many electronic bands closely spaced in energy which the multilayer graphene systems realize over a wide range of angles. Physically, this enhancement effect mainly depends on the fact that the energy difference between an initial/final state and a virtual state is small and that there is a finite interband-matrix element between the two bands.

Thermal fluctuation or phonon displacement effects are generally suppressed for the virtual states as they are unoccupied. We do agree with the Referee, as mentioned in the previous response, that thermal fluctuations in the states involved in real transitions can become important for low frequency case ($\sim 1\text{meV}$). Apart from that, our results are very robust to realistic temperatures that shift current experiments in twisted graphene systems are performed (\sim about 10K-20K, please see APS results from the Koppens group).

Following Referee's question we added a clarification that the enhancement of shift current due to virtual transition is present at all dopings and included a new supplemental figure (an analog of Fig. 4c, but for other dopings from Fig. 4b) to demonstrate this point.

Changes to the Manuscript: Modified a discussion regarding the filling range over which virtual processes enhance shift current and added a new supplemental figure.

The value in the proposed scheme is in the large number of tunable parameters. The authors chose to investigate only 1 out of 5 possible interlayer twist angles. They may wish to comment on the possibility of obtaining even stronger enhancements by tuning all twist angles independently.

We thank the referee for the positive assessment. In this manuscript we focused on the alternating angle stacking of the multilayer graphene as then its electronic structure (in the absence of external displacement field) can be brought to a block diagonal form (Phys. Rev. B 100, 085109) as well as of its experimental relevance. The Referee is correct that indeed other twisting schemes could exist as proposed in literature. We believe however that an exhaustive search of the parameter space is outside the scope of the current manuscript as we are focusing on identifying a new physical mechanism. In future work, we plan to explore this additional degree of tunability using the FS metric criterion and the stability arguments from the paper Referee shared with us to characterize the parameter space for enhanced shift current in multilayer graphene systems. In light of Referee's comment, we added further discussion on this additional degree of tunability in the manuscript.

Changes to Manuscript: added a discussion about tunability of the devices with different twist angles between multilayers.

Reviewer #2 (Remarks to the Author):

This work claims to propose a general design principle for strong shift-current materials based on enhancing virtual transitions.

This claim sounds very interesting but is not supported enough by data. Also, the authors need to support their claim of proposing design principles with a more thorough and concrete analysis. The authors present many calculation results that show significant contributions originating from virtual transitions. However, I do not see a systematic control of the virtual-transition contributions. What is the dominant variable that controls the virtual part, while not affecting the direct-transition part? Moreover, while the velocity-gauge formula Eq. (3) contains virtual transitions, the same expression in the length gauge does not contain any virtual transition.

We would like to thank the referee for their critical evaluation of our work. We hope that our reply convinces the Referee of the merits of our manuscript.

Firstly we want to address the referee's comments regarding the length and velocity-gauge calculations. Calculations of shift current, or in fact any other response, can be carried out in either of the two gauges with both gauges being formally identical (see our Appendix 1, and Refs. 4,5 for explicit discussion). Both gauges however offer different technical advantages (from the point of view of the calculation) and thus are employed by authors interchangeably depending on the relevant situation. Similarly certain physical mechanisms, one of them being the enhancement of shift current thatift current by virtual transitions - the subject of our work, may be more apparent in one picture than another. In length gauge formulation that involves only initial and final states explicitly, presence of additional electronic bands in the spectrum modifies the quantum geometry of the system altering the structure of the initial and final quantum states. In velocity-gauge, that is derived using sum rules (See also Ref. 4), this quantum geometry contribution stemming from virtual bands becomes directly apparent and was the reason for our manuscript. We stress that if equivalent calculation would have been done in length gauge, in the context of either the SSH toy-model or the multilayer graphene, then upon hybridization of the energy blocks in the Hamiltonian, which alters the structure of wavefunctions, similarly an enhancement to total shift current would have been seen. The microscopic origins of it, however, would not have been as readily apparent.

A similar quantity, whose properties become more apparent in one method of calculation than another, is the Berry curvature which is one of the most studied QG quantities. It can be evaluated in two ways: (1) From the curl of an intraband Berry connection (Eq. 1.1 in Ref. [1]) (2) Also, as a sum of a product of velocity operators over all other bands (Eq. 1.3 in Ref.[1]). On the surface level, the first approach might look completely unrelated to the number and properties of other bands, but it arises because the given band cannot be simply described by a single state in a given unit cell. For a given band, the variation of the Bloch state in 'k' space is indirectly related to the dimensionality of the Hilbert space of a unit cell which in turn is connected to the multiband aspect of electronic properties. A large number of bands or degrees of freedom in the system allows for the possibility of a large variation in eigenstates. Another important point to note in the second approach is that the magnitude of Berry curvature increases as the separation of the given band from other bands decreases. In general, the more closely spaced the bands are, the larger the magnitude of most quantum geometric quantities like the Fubini-Study metric is as we argued in the paper.

With this analogy to Berry curvature, let us comment in more detail on the length-gauge versus momentum-gauge calculation of the shift current. The shift-current integrand (i.e. the object that becomes integrated over momentum space to yield shift current) which is an interband quantity can be expressed either just using the two bands directly involved in transition or by summing over all other bands. In the length gauge picture, the shift current integrand is defined using a generalized derivative which requires only two bands for evaluation. However, this quantity depends on interband and intraband connections and their derivatives which are inherently dictated by the multiband nature of bands. As shown in Ref. [4], Eq. 24, this generalized derivative can be evaluated by summing over all other bands. These kinds of sum rules make the role of virtual bands more transparent and are also easy to handle numerically as

they can all be expressed in terms of velocity matrices. These sum rules are also often employed to establish the equivalence of length gauge and velocity gauge picture.

We also want to draw the Referee's attention to the fact that our work is not the only work that focuses on the explicit role virtual transitions can play in determining the shift-current response. In fact, Ref.[3] which relates shift-current integrand to a Hermitian connection on a Riemannian manifold highlight this aspect on page 3 below Eq. 8 where the contribution to virtual transitions is shown to give rise to a non-trivial torsion to this connection and hence leading to an additional possibility of enhancing shift-current. For a two-band picture for example, if the hamiltonian is linear, any 'aaa' component would be zero as only this non-trivial torsion which can contribute to shift-current conductivity is zero due to the absence of virtual transitions.

To the best of our knowledge, no other work has explored the shift-current photovoltaic design principles based on these virtual transitions. We demonstrated this aspect with different tuning knobs (that are explicitly experimentally tunable!) in our work. First, we explore how the displacement field which controls the interband mixing (indirectly related to enhanced interband velocity matrix elements) for virtual bands affects the shift-current conductivity. These results are shown in Fig. 1 where increasing the displacement field leads to a significantly enhanced effect. Next, we also explore how these interband mixing and energy level spacing depends on the number of layers and twist angle. While none of these knobs explicitly turns off virtual transitions compared to direct transitions, as the referee asks, they do change the overall importance of the virtual transitions by adding more electronic states and modifying the quantum geometry. Lastly, our idea of these generalized design principles is also demonstrated using a toy model which illustrates the role of virtual transitions in a very lucid manner.

In conclusion, I do not recommend this paper for publication unless 1) the authors clearly present a tuning knob for the virtual-transition parts with data and 2) how can the virtual transition takes a distinguished role while the length-gauge formula does not distinguish virtual and direct transitions.

We hope that our answer explains to the Referee what we meant by virtual transitions and demonstrates where in length gauge this enhancement by virtual transitions is present. We apologize if it came across as suggesting that one calculational method (velocity-gauge formalism) is physically any different from the other calculation (length-gauge formalism). In light of Referee's concerns, we added a clarifying paragraph in the manuscript that explicitly comments on the length versus velocity gauge calculations and how presence of additional bands enabling virtual transitions manifests in both formalisms. We also emphasize the role of the displacement field in promoting these additional virtual transitions as a tuning knob that the Referee asked for.

Changes to the Manuscript: Added discussion on the length gauge versus velocity-gauge formalism and how they relate to our claims.

References:

1. Xiao, Di, Ming-Che Chang, and Qian Niu. "Berry phase effects on electronic properties." *Reviews of modern physics* 82.3 (2010): 1959.
2. Ahn, Junyeong, Guang-Yu Guo, Naoto Nagaosa, and Ashvin Vishwanath. "Riemannian geometry of resonant optical responses." *Nature Physics* 18, no. 3 (2022): 290-295.
3. Sipe, J. E., and A. I. Shkrebtii. "Second-order optical response in semiconductors." *Physical Review B* 61, no. 8 (2000): 5337.
4. Cook, A. M., Fregoso, B. M., De Juan, F., Coh, S., and Moore, J. E. "Design principles for shift current photovoltaics" *Nature communications* 8, 1 (2017)
5. Parker, D. E., Morimoto, T., Orenstein, J., and Moore, J. E. "Diagrammatic approach to nonlinear optical response with application to weyl semimetals", *Phys. Rev. B* 99, 045121 (2019).

Reviewer #3 (Remarks to the Author):

In the manuscript, the authors discussed a mechanism that exploits virtual transitions in multiband systems to enhance shift current response. They found that the enhanced shift current response is achieved through increasing the number of possible virtual transitions and maximizing Wannier function spread that is related to Fubini-Study (FS) metric (or quantum geometry). They demonstrated the viability of the mechanism using a one-dimensional multi-chain Rice-Mele model. Furthermore, they proposed that a twisted multi-layer graphene with a proper twist angle exhibits the large shift current response.

As the authors mentioned in the paper, there is a direct connection between shift current response and FS metric in one dimension, and FS metric provides a lower bound for the spread of Wannier function. I have a few questions here.

- Could the authors show a similar connection in two dimensions?

We thank the referee for this question. The results of Appendix 2 demonstrate the relation between shift current and the FS metric in both 1D and 2D - we apologize if we have not emphasized this point more strongly in the manuscript. We updated the manuscript with new discussion. Indeed the search for the optimal twist angles that give rise to maximal shift current in multilayer graphene systems relies on identifying which twist angles optimize the FS metric (please see Fig. S5).

Changes to Manuscript: Clarified in the manuscript applicability of our results to two dimensions.

- Related to the first question: dose large FS metric generally tend to exhibit large shift current response?

We thank the Referee for this excellent question. Indeed typically yes - large FS generally points towards strong quantum geometric features of electronic bands which in general dictate the behavior of quantities like shift current response.

The shift current expression however, see Eg. S22 that explicitly involves the FS metric, relies not only on the metric but also on other quantities such as energy separation between bands for example. Ideally, a material candidate with large FS metric and many closely spaced electronic bands is the most suitable for studying shift current response.

Overall, the paper is clearly and logically structured. I believe that the results will be beneficial for people who search for materials that exhibit large shift current response. Therefore, I recommend this paper to be published in Communications Physics.

We would like to thank the referee for their positive feedback on our work, the recommendation to publish the manuscript in Communications Physics, and we appreciate their willingness to go through our manuscript and provide a review.

REVIEWERS' COMMENTS:

Reviewer #1 (Remarks to the Author):

The authors have responded satisfactorily to all the comments. I recommend the paper for publication.

Reviewer #2 (Remarks to the Author):

The authors addressed all my concerns satisfactorily. Now, I understand that they already presented data supporting that the enhanced response originates from virtual transitions. They also clarified the confusion arising from different descriptions based on velocity and length gauge approaches. Thus, I recommend this paper for publication in Communications Physics.